# Comprehensive analysis platform to understand, remedy, and eliminate amyotrophic lateral sclerosis (CAPTURE ALS): Study protocol for a Canadian multicenter, multimodal, longitudinal observational study

Natalie Saunders[1], Claire Magnussen[1], HyungMo Kang[1]*, Mathieu Blais[2], Harpreet Bhinder[3], Gerald Pfeffer[4], Shelagh K. Genuis[3], Liziane Bouvier[5], Tanushka Anand[3], Rida Abou-Haidar[1], Agessandro Abrahao[6], Marie-Noëlle Boivin[1], Robert Bowser[7], Tania Bubela[8], Julia Chiappini[1], Samir Das[1], Avnit Dhanoa[3], Nicolas Dupré[2,9], Alan Evans[1], Nicolas Ferry[1], Yvonne Frater[6], Angela Genge[1], Simon J. Graham[6], Russell Greiner[10], Yasser Iturria Medina[1], Wendy S. Johnston[3], Kelvin E. Jones[11], Jason Karamchandani[1], Jasna Kriz[12], Westerly Luth[3], Geneviève Matte[13], Ekaterina Rogaeva[14], Janice Robertson[14], Peter Seres[15], Fred Tam[6], David Taylor[16], Clémence Tremblay-Desbiens[2], Christine Vande Velde[13], Yana Yunusova[17], Lorne Zinman[6], Sanjay Kalra[3]

**1** The Neuro-Montreal Neurological Institute- Hospital, McGill University, Montreal, Québec, Canada, **2** Neuroscience Axis, CHU de Québec – Université Laval, Québec City, Québec, Canada, **3** Department of Medicine, Division of Neurology, Faculty of Medicine and Dentistry, University of Alberta, Edmonton, Alberta, Canada, **4** Hotchkiss Brain Institute, University of Calgary, Calgary, Alberta, Canada, **5** School of Communication Sciences and Disorders, McGill University, Montreal, Québec, Canada, **6** Sunnybrook Health Sciences Centre, University of Toronto, Toronto, Ontario, Canada, **7** Department of Translational Neuroscience, Barrow Neurological Institute, Pheonix, Arizona, United States of America, **8** Faculty of Health Sciences, Simon Fraser University, Burnaby, British Columbia, Canada, **9** Department of Medicine, Faculty of Medicine, Université Laval, Québec City, Québec, Canada, **10** Department of Computing Science, Faculty of Science, Alberta Machine Intelligence Institute, University of Alberta, Edmonton, Alberta, Canada, **11** Faculty of Kinesiology, Sport and Recreation, University of Alberta, Edmonton, Alberta, Canada, **12** Department of Psychiatry and Neurosciences, Faculty of Medicine, Université Laval, Québec City, Québec, Canada, **13** Department of Neurosciences, Université de Montréal, Montreal, Québec, Canada, **14** Tanz Centre for Research in Neurodegenerative Disease, University of Toronto, Toronto, Ontario, Canada, **15** Department of Radiology and Diagnostic Imaging, University of Alberta, Edmonton, Alberta, Canada, **16** ALS Society of Canada, Toronto, Canada, **17** Department of Speech-Language Pathology, University of Toronto, Toronto, Ontario, Canada

* hyungmo.kang@mcgill.ca

## Abstract

### Background

The marked heterogeneity of Amyotrophic Lateral Sclerosis (ALS) combined with a lack of biomarkers are key contributing factors to the lack of disease-modifying treatments. The Comprehensive Analysis Platform to Understand Remedy and Eliminate ALS (CAPTURE ALS) is a Canadian platform designed to create the most comprehensive picture of people living with ALS with the objective of facilitating ALS research initiatives worldwide.

**Data availability statement:** Standard operating procedures (SOPs) and selected case report forms are provided in supporting information. There is no data published in this manuscript, except for recruitment numbers. The CAPTURE ALS protocol, SOPs, and information on how to access data and biosamples from the CAPTURE ALS database are published on http://capture-als.ca. Data and biosamples can be requested through open science access principles through this URL: https://cbigr-open.loris.ca/. Data will start to be accessible beginning Q4 of 2025.

**Funding:** The CAPTURE ALS platform has been made possible with the financial support of Health Canada, through the Canada Brain Research Fund, an innovative partnership between the Government of Canada (through Health Canada) and Brain Canada (https://braincanada.ca/), and of the ALS Society of Canada (https://als.ca/), Alnylam Pharmaceuticals (https://www.alnylam.com/), Regeneron Pharmaceuticals (https://www.regeneron.com/), the Canadian Institutes of Health Research (CIHR) (https://cihr-irsc.gc.ca/e/193.html) (SK, YM), and the Chris Snow/Calgary Flames Foundation (SK). The views expressed herein do not necessarily represent the views of the Minister of Health or the Government of Canada. The onboarding of CHUM (Centre hospitalier de l'université de Montreal) has been made possible with financial support of the ALS Society of Quebec (https://sla-quebec.ca/en/) and ALS Action Canada (https://www.alsactioncanada.org/) (GM). The funders had no role in study design, data collection and analysis, decision to publish, or preparation of the manuscript. There was no additional external funding received for this study.

**Competing interests:** The authors have declared that no competing interests exist.

## Objectives

The main aims of CAPTURE ALS include: (1) to characterize ALS and healthy controls with biosamples and data in order to provide the most comprehensive picture of individuals living with ALS to date; (2) to create a de-identified database and biosample repository linked to detailed clinical information; and (3) to develop and implement an inclusive and transparent participant engagement strategy to be active throughout all stages of CAPTURE ALS.

## Methods/Results

CAPTURE ALS is a prospective, multicenter, observational, longitudinal study. People living with ALS, or a related disease and healthy controls undergo a harmonized protocol including the collection of detailed clinical information, neurological and cognitive examination, speech recording, advanced magnetic resonance imaging, and biosampling. Data and samples are stored in a biobank operating under an open science governance framework. An inclusive and transparent participant engagement strategy was designed and implemented throughout all stages of CAPTURE ALS. Four sites are operating in the consortium with a fifth being onboarded. The target enrollment is 120 affected participants and 50 controls, with the first participant visit having occurred in March 2022. Recruitment is ongoing.

## Discussion

CAPTURE ALS is a scalable clinical research platform that connects scientists and patients to facilitate efficient translational research. The unique and deeply phenotyped data and biosamples are a global resource towards the development of biomarkers and understanding ALS biology. This study is registered at clinicaltrials.gov (NCT: NCT05204017).

## Introduction

### Background and rationale

Amyotrophic lateral sclerosis (ALS) is a highly heterogenous disease, with a clinical presentation and progression varying significantly between patients. Marked variability is seen with respect to age of onset, site of onset, relative degree of upper (UMN) and lower motor neuron (LMN) involvement, cognitive impairment, and genetic abnormalities. The presentation is complicated by ALS' disease spectrum with frontotemporal dementia (FTD) and its clinical overlap with primary lateral sclerosis (PLS) and progressive muscular atrophy (PMA) [1–3]. Phenotypic heterogeneity in ALS is likely underpinned by different pathogenic mechanisms [4,5], perhaps contributing to why so many ALS clinical trials have failed [6].

There is a general consensus in the field that a precision medicine approach, which considers an individual's genotype and phenotype, will be required to

effectively treat ALS [7,8]. To this end, groups are exploring methods to stratify patients into homogeneous subgroups. Genetic stratification in ALS has provided specific targets for drug development, spawning clinical trials with gene therapy approaches. Tofersen, a recently FDA-approved antisense oligonucleotide designed to downregulate SOD1 [9,10] is an example of such an approach that has shown benefit in the specific group of patients with a mutation in the SOD1 gene. However, precision medicine becomes more complex with non-genetic forms of ALS, which account for approximately 90% of cases of ALS [11].

Major infrastructures have been developed to support ALS clinical trials, share clinical datasets, and establish bio-specimen repositories in order to advance the understanding of ALS. In a recent review, 22 registries and collaborative efforts were noted across North America, South America, Europe, and Asia [12]. Shared clinical databases enhance the coordination of resources and data analyses while biorepositories help ensure standardization and quality controls. These infrastructures provide critical resources to the research community. Examples of large ALS platforms collecting biological materials from people with ALS include Answer ALS, AMBRoSIA and CReATE [13]. These platforms also differ in their objectives, scope and protocols.

A major challenge in ALS research is to obtain comprehensive and longitudinal characterization of disease phenotype beyond basic clinical characteristics such as age, sex, time from symptom onset, region of onset, ALS Functional Rating Scale-Revised (ALSFRS-R) score, and respiratory vital capacity measurements. Linking genetic data and other biosamples to such thorough characterization would also be a key asset, though it is an additional challenge in ALS research. Meeting these challenges would allow a thorough, quantitative examination of the phenotypic and temporal heterogeneity in ALS, and would promote advances in biomarker discovery.

The Comprehensive Analysis Platform to Understand, Remedy and Eliminate ALS (CAPTURE ALS), is a Canadian open science research platform that was designed in response to the need to robustly characterize ALS heterogeneity, understand ALS biology and develop biomarkers. The study protocol presented here was developed largely on that used by the Canadian ALS Neuroimaging Consortium (CALSNIC) [14]. CALSNIC, which was founded by the same group, yielded two observational studies (CALSNIC-1 and CALSNIC-2) that acquired multimodal neuroimaging, neurological, neuropsychological, and speech data from over 250 ALS patients and 200 healthy controls. As the first multi-center, prospective, longitudinal neuroimaging study in ALS, CALSNIC used these unique datasets to develop and evaluate advanced MRI-based biomarkers that delineate biological heterogeneity, track disease progression and predict survival [15–20]. CAPTURE ALS expanded upon the CALSNIC study protocol to yield the most comprehensive clinical dataset linked to biological samples ever attempted in the field. Added in the CAPTURE ALS protocol is the collection of biosamples, for which standard operating procedures (SOPs) were developed under the guidance of and support of consortia preceding CAPTURE ALS, in particular the Northeast Amyotrophic Lateral Sclerosis Consortium (NEALS). Consultations with collaborators and other consortia were made to optimize CAPTURE ALS' interoperability with materials of other consortia. To further ensure the protocol's implementation, CAPTURE ALS incorporates the voices of people affected by ALS, and therefore created a Participant Partner Advisory Council (PPAC), which advises on the protocol and any proposed changes.

The datasets and biospecimens acquired through CAPTURE ALS are housed at the Clinical Biological Imaging and Genetic (C-BIG) Repository [21], whose technical, physical, ethical and legal framework was leveraged to permit sharing through open science with the global research community.

## Objectives

CAPTURE ALS is a long-term research platform, with the overarching objective of advancing the understanding of ALS and defining its clinical heterogeneity. By making datasets available for researchers globally, the platform will facilitate investigations into the fundamental molecular alterations underlying ALS, foster the discovery of new biomarkers, and allow tissue- and fluid-based translational research to develop new treatments for ALS and related disorders. This will be achieved using a multidisciplinary approach. There are three specific aims:

1. Characterize people affected by ALS and related motor neuron diseases (MNDs), and healthy controls with deep clinical phenotyping, neuroimaging, speech analyses, whole genome sequencing, and multisource analyses.

2. Develop a de-identified database and biosample repository linked to detailed clinical information, that will be a resource for the Canadian and international scientific communities.

3. Develop and implement an inclusive, transparent, structured patient engagement strategy to be active throughout all stages of CAPTURE ALS.

## Materials and methods

### Study design

The CAPTURE ALS research platform operates a prospective, longitudinal study enrolling people living with ALS (or related diseases) and healthy controls. It is a multi-center study that is currently recruiting at 4 Canadian ALS multidisciplinary clinics at the University of Alberta (Edmonton, Alberta), McGill University (Montreal, Quebec), Université Laval (Quebec City, Quebec) and the University of Toronto (Toronto, Ontario). A fifth site at the Centre hospitalier de l'université de Montréal is being onboarded. CAPTURE ALS follows best practices for Patient Oriented Research, as outlined by CIHR/SPOR [22,23].

### Participant engagement

Patient engagement can facilitate recruitment and retention [24,25], but more importantly, it is the key to improving the relevance, feasibility, impact, and efficiency of research and can improve patient outcomes [26–28]. CAPTURE ALS is designed to incorporate the voices of people affected by ALS, including those living with ALS, their loved ones, friends, and communities. The PPAC holds regular meetings, providing ongoing input on study needs and challenges including input on research priorities and strategies for participant recruitment. One person living with ALS and one caregiver from each CAPTURE ALS site, as well as 1–3 representatives from the ALS Canada Ambassadors Program participate as PPAC members. Members have an anticipated term of 1–2 years with opportunities for longer participation. In addition to the PPAC, CAPTURE ALS' participant engagement strategy includes a "Participant Experience Questionnaire". This questionnaire is included in the study protocol and schedule of procedures for all participants. At the end of the last study visit, participants have the option to complete an online questionnaire to provide the research team with feedback on their overall experience as CAPTURE ALS research participants. The aim is to better understand research participation from the perspective of participants, and to explore how this experience can be improved for future ALS studies.

### Study population

There are two main study populations in CAPTURE ALS. The first is the affected group that includes people living with ALS, a related MND (such as PLS or PMA), and presymptomatic individuals with a known ALS mutation. Affected participants are permitted to be enrolled in another clinical trial in parallel to their participation in CAPTURE ALS. Few inclusion/exclusion criteria are employed to ensure that the true heterogeneity in the ALS population is captured. Since the goal is to understand the different presentations of the disease across a spectrum, individuals with ALS meeting any designation of the revised El Escorial criteria [29], are eligible to participate. The second group is healthy controls. The inclusion and exclusion criteria for patients and controls are outlined under study procedures. In addition to affected participants and controls, caregivers have the opportunity to participate in CAPTURE ALS by consenting to complete behavioral assessment questionnaires.

### Ethics

CAPTURE ALS is conducted in accordance with the Good Clinical Practice guidelines of the International Council for Harmonisation [30] and the Tri-Council Policy Statement: Ethical Conduct for Research Involving Humans (TCPS2) [31].

Each site has received research ethics board (REB) approval from their respective institutional ethics committees to recruit participants. The consent provided by participants enables broad secondary use of their de-identified and coded data/materials for research on ALS, related diseases, and other neurological diseases as well as comparative research on other human diseases and conditions by both academia and industry in Canada and globally.

## Status and timeline of the study

CAPTURE ALS launched at the University of Alberta with the first study visit of the first affected participant in March 2022. Recruitment at the other three sites initiated over the course of the following year. Recruitment is ongoing. This is a long-term research platform with no restrictions on sample size or timeframe beyond those permitted by available funds. The current funding permits recruitment of 120 affected participants and 50 healthy control participants. Sites will continue to submit the annual renewal for the study at their REB until the target recruitment numbers are reached, or as long as funds are available to continue study visits. Recruitment is expected to be completed in December 2026, and data collection in December 2027.

## Study procedure

**Recruitment and consent.** CAPTURE ALS is a research platform with the objective of making multimodal, longitudinal data and materials from ALS patients and controls readily available to scientists for their individual research. As such, no formal sample size calculations were performed. CAPTURE ALS is currently funded to recruit 120 affected participants and 50 healthy control participants.

Affected participants and control participants are primarily recruited through the ALS multidisciplinary clinics at each of the participating sites where they are referred to the research team by their clinical care team. Control participants are also recruited through advertisements.

The informed consent process is managed by a dedicated CAPTURE ALS research coordinator at each site. The consent is obtained in-person, and all participants must provide written informed consent prior to performing any study procedures. Participation is entirely voluntary and may be revoked by the participant at any time without having an impact on their clinical care.

## Eligibility criteria

### Inclusion criteria

### Affected participant

1. Has ALS, classified as definite, probable, laboratory-supported probable, or possible ALS according to the revised El Escorial criteria, a related neurodegenerative disorder (including ALS-FTD, PLS, PMA), or is an asymptomatic individual with a known ALS mutation.

2. Be of the age of majority in their province of residence/treatment.

3. Has the cognitive capacity to provide informed consent.

4. Has proficiency in English or French to understand study instructions and respond to questionnaires.

### Control participant

1. Be between the ages of 40−80 years unless age-matched to a patient (+/-3 years) who is already enrolled.

2. Be of the age of majority in their province of residence/treatment.

3. Has the cognitive capacity to provide informed consent.

4. Has proficiency in English or French to understand study instructions and respond to questionnaires.

***Exclusion criteria.***

***Affected participant***

There are no exclusion criteria for patients except for the following criterion for those undergoing an optional Lumbar Puncture (LP):

 1.Taking an anticoagulant, or is at risk of increased or uncontrolled LP-related bleeding due to factors including, but are not limited to, anatomical factors at or near the LP site (e.g., vascular abnormalities, neoplasms, or other abnormalities) or abnormal platelet or coagulation test values at screening. The use of aspirin is permitted at the investigator's discretion.

***Control participant***

1. A history of neurological disease, including Central Nervous System disease (e.g., stroke, head injury, epilepsy) or Peripheral Nervous System disease (e.g., neuropathy, myopathy).

2. A history of psychiatric disease (e.g., depression) that is clinically diagnosed and/or there is the current use of psychiatric medications (e.g., antidepressants) for an indication of a psychiatric disease.

3. Ineligible for an MRI due to a pacemaker or other contraindication according to local MRI policies of the study centre.

4. Significant claustrophobia that would prohibit an MRI.

## Data collection

All study investigators, research coordinators, and other study personnel are required to undergo documented training on the study protocol and have an up-to-date Good Clinical Practice certificate. CAPTURE ALS SOPs were created for specific tasks in the protocol. Study personnel delegated to perform these tasks must have reviewed the relevant SOP and have it documented on the training log prior to performing the task. A full list of the SOPs can be found in Supporting Information.

All participants undergo deep clinical phenotyping, neurocognitive testing, multimodal neuroimaging, speech analysis, and multisource biospecimen collection and analyses. These assessments are described under Primary Outcomes. Affected participants have study visits at Month 0 (baseline), Month 4, Month 8, and Month 12. Control participants have two study visits: Month 0 (baseline) and Month 8. Both affected and control participants must pass a screening visit, which can be combined with the baseline visit to reduce participant burden. Study procedures performed at each visit are summarized in Table 1 for affected participants and Table 2 for control participants. Compared to affected participants, control participants do not undergo neurological examinations, ALSFRS-R, Forced Vital Capacity (FVC), Frontal Systems Behavior Scale (FrSBe) and the ALS Assessment Questionnaire Short Form (ALSAQ-5).

## Primary outcomes

The following clinical assessments are performed to undergo deep phenotyping

**Demographic information and disease onset.** Demographic and disease onset information are collected (S1 Appendix). Demographic information includes and is not limited to age at the screen visit, sex at birth, race, hand preference, occupation status, education, where the participant was primarily raised, communication methods, and past/present clinical trial participation information.

Considering the significant individual heterogeneity in symptom onset/disease progression in the ALS population and various operational definitions of disease onset in the field, its consistent definition across the entire sample size is crucial

**Table 1. Schedule of study procedures for affected participants.**

| Affected Participants | Visit 1 | Visit 2 | Visit 3 | Visit 4 | Visit 5 | Long Term Follow Up |
|---|---|---|---|---|---|---|
| | −1 | 0 month | 4 months | 8 months | 12 months | Every 6 months from Visit 5 |
| | Screening | Baseline | | | | |
| Informed Consent | X | | | | | |
| Onset Information | X | | | | | |
| Study Status | X | X | X | X | X | X |
| Medical Evaluations | X | X | X | X | X | |
| Family History and Genetic Testing results | | X | X | X | X | X |
| ALSFRS-R | | X | X | X | X | X |
| Forced Vital Capacity (FVC) | | X | X | X | X | X |
| Finger/Toe tapping | | X | X | X | X | |
| Saccades | | X | X | X | X | |
| Speech Testing | | X | X | X | X | |
| Neurological Exam* | | X | X | X | X | |
| El Escorial | | X | X | X | X | |
| MRI | | X | X | X | X | |
| Montreal Cognitive Assessment (MoCA) | X (V8.1) | | | | | |
| Edinburgh Cognitive and Behavioural ALS Screen (ECAS) | | X (VA) | X (VB) | X (VC) | X (VA) | |
| Neurocognitive battery** | | X | X | X | X | |
| Participant Experience Questionnaire | | | | | X | |
| **Patient Reported Outcome Measures** | | | | | | |
| World Health Organization Quality of Life-BREF Scale (WHOQOL-BREF) | | X | X | X | X | |
| The Amyotrophic Lateral Sclerosis Assessment Questionnaire Short Form (ALSAQ-5) | | X | X | X | X | |
| **Caregiver Questionnaires** | | | | | | |
| ECAS Behavior Screen | | X | X | X | X | |
| FrSBe (Family-Rating form) | | X | | X | | |
| Participant Experience Questionnaire | | | | | X | |
| **Biosampling** | | | | | | |
| Blood Collection | | PMBC, Serum, Plasma, DNA, RNA | Serum, Plasma | Serum, Plasma | Serum, Plasma | |
| Cerebrospinal Fluid (CSF) (optional) | | X | X | X | X | |

*See S1 Table for details of the neurological examination features.

**See S2 Table for details of the neurocognitive battery for affected participants.

in obtaining the standardized data [32]. In this study, symptom onset date is defined as the date when limb weakness, speech/swallowing difficulties, dyspnea, or generalized weakness was first noticed. Site of onset of progressive weakness is collected (limb, bulbar, respiratory), and more specific information about onset location is collected if the symptoms began in patient's limb (upper/lower, left/right). In affected participants where the first symptom is not weakness (e.g., fasciculations, atrophy), a separate "first symptom" date is collected.

**Study status.** Every study visit documents a participant's status in the study (either Active, Withdrawn, Lost to Follow-Up, or Deceased). If the participant is deceased, date and cause of death (Medical Assistance in Dying (MAiD) or other

**Table 2. Schedule of study procedures for control participants.**

| Control Participants | Visit 1 | Visit 2 | Visit 3 |
|---|---|---|---|
| | −1 | 0 months | 8 months |
| | Screening | Baseline | |
| Informed Consent | X | | |
| Study Status | X | X | X |
| Medical Evaluations | X | X | X |
| Family History and Genetic Testing results | | X | X |
| Finger/Toe tapping | | X | X |
| Saccades | | X | X |
| Speech Testing | | X | X |
| MRI scan | | X | X |
| Montreal Cognitive Assessment (MoCA) | X (V8.1) | | |
| Edinburgh Cognitive and Behavioural ALS Screen (ECAS) | | X (VA) | X (VC) |
| Neurocognitive battery*** | | X | X |
| Participant Experience Questionnaire | | | X |
| **Participant Reported Outcome Measures** | | | |
| World Health Organization Quality of Life-BREF Scale (WHOQOL-BREF) | | X | X |
| **Biosampling** | | | |
| Blood Collection | | Serum, Plasma, RNA | Serum, Plasma |

***See S3 Table for details of the neurocognitive battery for control participants.

causes, if known) are collected (S2 Appendix). For affected participants, the date of the following medical interventions is documented: Percutaneous Endoscopic Gastronomy (PEG) insertion, Bilevel Positive Airway Pressure (BiPAP), Permanently Assisted Ventilation (PAV, defined as greater than 22 hours/day), Tracheostomy.

**Medications and medical history.** A complete list of participant's medications, including but not limited to Riluzole, Edaravone, Albrioza (AMX0035), psychiatric medications, and baclofen are collected (S3 Appendix). Other health-related information such as smoking history, current alcohol and recreational drug use is collected to account for their potential effects on forced vital capacity (FVC), brain structure, performance in cognitive assessment, and biosample measurements.

Participant's medical conditions and history are collected at the screening visit and updated every study visit. To remain consistent with CALSNIC, a more targeted medical history is collected including history of brain trauma (defined as an injury severe enough to result in loss of consciousness), stroke, cognitive/learning issues, epilepsy, other neurological conditions, depression, psychiatric conditions, malignancy, diabetes, hypertension and other significant medical conditions.

**Family history and genetic testing.** Information on family history of ALS, other MNDs, other neurological disorders, dementia, and psychiatric illnesses is collected (S4 Appendix). Available genetic testing results through standard of care are collected including gene(s) involved and identified pathogenic variants or repeat length.

**ALSFRS-R [affected participants only].** The revised ALSFRS is the gold standard used to assess the degree of functional impairment in patients with ALS and is used as a global measure of disease status and progression [33]. It measures activities of daily living and global function across four domains: bulbar, fine motor, gross motor, and respiratory. Despite its wide use, the scale is often inconsistently used [34,35]. To ensure consistency and reproducibility, the ALSFRS-R is administered to affected participants by a trained evaluator at each study visit, following the CAPTURE ALS ALSFRS-R SOP (S5 Appendix). The SOP is adapted from the European Network for the Cure of ALS (ENCALS) SOP,

with some differences on scoring: 1) Swallowing (item 3): ENCALS indicates that a weight loss of >10% is required to score 3, however similar to the NEALS SOP, it is not specified for the CAPTURE ALS SOP; 2) Dyspnea (item 10): pre-existing respiratory issues such as asthma, do not affect the score to only capture ALS symptomology. If a patient is using a form of non-invasive ventilation (NIV), score is based on how the patient feels when not using NIV, similar to the NEALS SOP; 3) Orthopnea (item 11), the use of Continuous Positive Airway Pressure therapy prior to ALS does not affect the score, however the ENCALS SOP indicates that dependency on non-invasive ventilation for most or all of the night results in a score of 0; 4) Respiratory insufficiency (item 12): scored based on a similar rationale as item 11.

**Forced vital capacity [affected participants only].** FVC, the maximal volume of air exhaled with maximally forced effort from a maximal inspiration [36], has been shown to be predictive of hypoventilation and survival. It is thus frequently used as an outcome measure in ALS trials [37]. In this study, FVC is documented at every study visit for affected participants as a measure of disease status. Standard of care measurements taken within 4 weeks of the study visit are recorded, otherwise it is obtained by a trained study team member. The best trial (L, % predicted), reliability of the test, position of the test as well as the participant's height, weight and breathing conditions are documented as well (S6 Appendix).

**Neurological evaluations.** Neurological assessments performed at every visit include finger and toe tapping, saccades and antisaccades, and a neurological exam (for affected participants only) as shown in the Neurological Evaluations Form (S7 Appendix).

Finger/Toe Tapping: Tapping speeds represent a measure of UMN dysfunction. Finger and toe tapping speeds are measured by counting the maximum number of taps performed in 10 seconds [38]. Two trials per limb are collected, alternating between sides to avoid possible fatigue.

Saccades/Antisaccades: Abnormal eye movements may provide insights into the pattern, pathogenesis, and severity of the disease [39–41]. A saccade is a volitional rapid eye movement designed to shift the fovea to objects of visual interest. An antisaccade tests the participant's ability to make volitional rapid eye movements in the direction opposite to the side where a stimulus is presented. At every study visit, 8 random saccade and antisaccade trials are performed.

Neurological Exam [affected participants only]: A complete neurological exam is done at every visit by a neurologist trained and delegated to the study protocol. The presence of fasciculations and atrophy is documented by body region. The severity of dysarthria is noted, as well as any facial or tongue weakness. Spasticity of each limb is assessed using the Modified Ashworth Scale [42]. Muscle strength is evaluated by manual muscle testing (MMT) of 19 muscles designated in the protocol: neck flexion and extension, 10 upper extremity muscles, and seven lower extremity muscles. These are graded using the Medical Research Council (MRC) scale [43]. Muscle stretch reflexes including jaw jerk, biceps, brachioradialis, triceps, quadriceps, triceps surae are quantified. Superficial abdominal reflexes, and pathologic reflexes including Hoffman and Babinski signs are documented. During the neurological examination and interaction with the patient, the presence of pseudobulbar affect is documented. Cerebellar dysfunction (finger-nose and heel-shin ataxia) as well as the gait (normal, abnormal, abnormal-spastic, abnormal-ataxic, unable to walk without aids) are documented.

Diagnosis: UMN and LMN involvement in the bulbar, cervical, thoracic and lumbar region are documented based on findings in the clinical exam and available standard-of-care needle electromyography results.

**Neurocognitive testing.** A full battery of neurocognitive testing is performed at Months 0 and 8 for all participants, and includes Montreal Cognitive Assessment (MoCA), Edinburgh Cognitive and Behavioral ALS Screen (ECAS), and a neurocognitive evaluation. For affected participants, only the ECAS and semantic fluency are conducted at Month 4 and 12. Patient/Participant Reported Outcome Measures (PROM) are administered at all visits except for screening visits. Neurocognitive assessments are performed according to the CAPTURE ALS Neurocognitive Testing SOP (S8 Appendix). S4 Table summarizes each instrument's purpose and administration details.

**Speech testing.** Most people with ALS, regardless of the site of onset, eventually experience bulbar motor impairment. ALS affects all the speech subsystems (articulation, resonance, phonation, respiration). Slowed speech rate and

impaired vocal quality are often among the first signs of bulbar impairment reported in ALS patients [44–47]. Alterations of the amplitude, speed and stability of the tongue, lips and jaw movements have also been reported [48–56], with an impact on articulation precision and rate. Acoustic and kinematic measures have been associated with the progression of bulbar disease, including measures of speech timing during passage reading, rate of syllable repetitions during oral dysdiadochokinesia (DDK), and tongue movements [54,57,58].

To study the association between bulbar motor impairment and other clinical abnormalities in ALS, participants undergo speech testing at every visit. Speech assessment tasks are recorded using an in-house, web browser-based, audio-video data collection tool. The speech testing protocol, which takes approximately 30 minutes to complete, includes the following: sustained phonation, pitch glides, diadochokinetic task, speech intelligibility, sentences and passage reading, picture description and open-ended narrative task. Analyses of the audio signal will extract a large set of acoustic features related to the temporal (variations in amplitude and frequency components over time), spectral (frequency content), and cepstral (periodic structure) characteristics of the speech signal. They will include measures related to articulation, phonation, resonance, and breathing (e.g., number of pauses, length of speech segments). Video data are used to extract kinematic features of the lips and jaw motions during speech, including range of motion, speed, velocity, and symmetry [59,60].

**Neuroimaging.** One of the major features that sets CAPTURE ALS apart from other international ALS initiatives is the collection of longitudinal brain imaging data using MRI on all participants. Brain imaging has provided insights into pathology and pathophysiological mechanisms in vivo in both symptomatic and presymptomatic individuals with ALS. The CAPTURE ALS neuroimaging protocol is derived and adapted from those used in CALSNIC-1 and CALSNIC-2 [14], as shown in Table 3. CALSNIC studies demonstrated the ability to monitor the progression of neurodegeneration using imaging metrics over an observation interval of 4−8 months [15,17,19], and that clinical heterogeneity (disease progression rate, UMN burden) is associated with MRI measures of cerebral pathology [16,18,20]. The harmonization across all 3 studies enables their integration and pooling of data. The multimodal protocol assesses structure (T1-weighted (T1w)), microstructure (diffusion tensor imaging (DTI)), neurochemistry (magnetic resonance spectroscopy (MRS)), resting state function (rs-fMRI), iron deposition (susceptibility weighted imaging (SWI)), blood flow (arterial spin labeling (ASL)), and co-existing pathology (fluid-attenuated inversion recovery (FLAIR)). All of these modalities have shown sensitivity to change in ALS compared to healthy controls in prior studies. Single voxel MRS of the motor cortex (MC) and prefrontal cortex (PFC) was performed in CALSNIC-1 but not done in CALSNIC-2; in CAPTURE ALS, the MC is probed, but the frontal voxel was not retained due to lower metabolite ratio precision from that region accompanied by a lack of robust cross-sectional and longitudinal change [61].

Sequence parameters are dependent on the MRI system (Siemens, GE, Philips) operating at the site. Some parameters are common across all systems. As examples, constants across respective sequences and MRI systems are image resolution, the number of DTI b-values and diffusion directions, and the number of volumes in rs-fMRI [14].

**Table 3. Imaging protocol across studies.**

|         | CALSNIC-1 | CALSNIC-2 | CAPTURE ALS |
|---------|-----------|-----------|-------------|
| 3D-T1   | •         | •         | •           |
| DTI     | •         | •         | •           |
| rs-fMRI | •         | •         | •           |
| MRS     | MC & PFC  |           | MC          |
| PD/T2   | •         | 3D T2     | •           |
| FLAIR   | 2D        | 3D        | 3D          |
| SWI     |           | •         | •           |
| ASL     |           |           | •           |

**Biosample collection.** Biosamples of participants of CAPTURE ALS are collected every study visit as per standard procedures at each CAPTURE ALS site. These include blood collection for serum, plasma and RNA in all participants (Tables 4 and 5). Affected participants provide a baseline blood sample for peripheral blood mononuclear cell (PBMC) isolation, and whole blood for DNA isolation. Cerebrospinal fluid (CSF) collection is optional. All biosamples collected through CAPTURE ALS are stored in the C-BIG Repository at the Montreal Neurological Institute (MNI) for future research purposes. Depending on the sample type, some biospecimens are processed at the collecting site before being shipped to C-BIG Repository according to the CAPTURE ALS Biosampling SOP (S9 Appendix).

**Biosample processing.** Blood collection and processing are performed by professional technicians. Isolation of serum and plasma is performed using standardized protocols at each site. Whole blood is collected into tubes containing RNA stabilizer for subsequent RNA isolation. Peripheral blood mononuclear cell isolation and whole blood collection for DNA isolation are performed at baseline visit for affected participants. See S5 Table for the summary of how collected biosamples are processed and stored for CAPTURE ALS.

## Secondary outcomes

**Whole genome sequencing.** Once shipped from each CAPTURE ALS site, DNA is isolated at the C-BIG Repository. DNA samples of affected participants from the CAPTURE ALS platform will undergo whole genome sequencing by Genome Quebec. Results from whole genome sequencing will be returned to the CAPTURE ALS database. Notably, the sequences and accompanying clinical data will be shared with the international ALS genetic consortium, Project MinE [62].

**Table 4. Schedule of blood collection for affected participants.**

| AFFECTED PARTICIPANTS | | |
|---|---|---|
| Time point | Collection | Total |
| Month 0 | Serum – 2 RTT (10 mL)<br>PBMC – 5 GTT (Heparin, 10 mL)<br>Plasma – 1 PTT ($K_2$EDTA, 10 mL)<br>DNA – 2 PTT ($K_2$EDTA, 10 mL)<br>RNA – 2 PAXgene RNA (2.5 mL) | vacutainers: 12<br>volume: 105 mL |
| Month 4 | Serum – 2 RTT (10 mL)<br>Plasma – 1 PTT ($K_2$EDTA, 10 mL) | vacutainers: 3<br>volume: 30 mL |
| Month 8 | Serum – 2 RTT (10 mL)<br>Plasma – 1 PTT ($K_2$EDTA, 10 mL) | vacutainers: 3<br>volume: 30 mL |
| Month 12 | Serum – 2 RTT (10 mL)<br>Plasma – 1 PTT ($K_2$EDTA, 10 mL) | vacutainers: 3<br>volume: 30 mL |

RTT (red top (glass) tube); GTT (green top tube); PTT (purple top tube)

**Table 5. Schedule of blood collection for control participants.**

| CONTROL PARTICIPANTS | | |
|---|---|---|
| Time point | Collection | Total |
| Month 0 | Serum – 2 RTT (10 mL)<br>Plasma – 1 PTT ($K_2$EDTA, 10 mL)<br>RNA – 2 PAXgene RNA (2.5 mL) | vacutainers: 5<br>volume: 35 mL |
| Month 8 | Serum – 2 RTT (10 mL)<br>Plasma – 1 PTT ($K_2$EDTA, 10 mL) | vacutainers: 3<br>volume: 30 mL |

**Neurofilament analyses.** Levels of neurofilament light chain (NF-L) and phosphorylated neurofilament heavy chain (pNF-H) will be analyzed in plasma samples in both affected and control participants using the Quanterix Simoa® NF assays. Quantification results will be made available in the CAPTURE ALS open science database.

**Performance of genome-wide DNA methylation.** The analysis of DNA methylation (DNAm) and DNAm-age will be performed in affected participants only. Results will be available for sharing through the CAPTURE ALS database.

## Data management

Each participant is assigned a study identification number at the time of informed consent by the research coordinator and is used on all source documents and database entries. The de-identified information collected in CAPTURE ALS is managed by the Longitudinal Online Research and Imaging System (LORIS). LORIS is a software platform used for managing data in multimodal studies. It was coded to meet the specific needs of CAPTURE ALS to manage the entire dataset, including the multimodal clinical assessments, imaging data, and biospecimen inventory details.

MRI scans are uploaded by the MRI technicians at each CAPTURE ALS site using a specialized MRI uploader that ensures data to be consistently captured and meet the standardized MRI protocol. Once an MRI scan is uploaded, it passes various parameter checks and flags any scan to the individual modality level to ensure MRI data are collected according to the predefined study protocols.

Clinical data entered in the LORIS database undergoes quality review at the central coordinating site (University of Alberta).

**Confidentiality and participant protection.** Participant's biological samples are labeled with their study ID before they are sent to the C-BIG Repository. Personal identifiers such as the participant's name, full date of birth, or hospital identification number will not be sent. No information that discloses the participant's identity will ever be shared.

## Progress

At the time of writing this paper, 118 participants had consented to participate in CAPTURE ALS (87 affected participants and 31 control participants). The mean age for male and female MND participants is 59 years old and 57 years old, respectively. For controls, it is 52 years old and 48 years old. In line with the literature [63], MND participants have a greater male-to-female ratio. The healthy controls exhibit a higher female-to-male ratio, a limitation that may be explained by male MND participants with female caregivers participating as healthy controls.

Regarding participant engagement, five people living with ALS and five caregivers are members of the PPAC. Since February 2021, 18 PPAC meetings have been held. Participant engagement webinars were initiated in March 2023; six have been held to date. The PPAC has contributed to governance documents, has provided their input on the study protocol and sub-studies, engagement webinar development and feedback surveys [64].

## Dissemination

The CAPTURE ALS database and biobank are located at the MNI's C-BIG Repository, which uses an open science framework. CAPTURE ALS is dedicated to offering an open science research platform. Those requesting CAPTURE ALS datasets and/or samples must have an ethics approved project aligned with the informed consent form of the CAPTURE ALS. There are increasing levels of access to data and materials available to researchers (https://captureals.ca):

1. **Open Access**: De-identified data with no risk of re-identification will be made publicly available. This will include summary data for example diagnosis, age and sex.

2. **Registered Access**: Researchers who register with CAPTURE ALS. These data include results of medical evaluations such as the ALSFRS-R, FVC and neurological examination.

3. **Controlled Access**: This includes all biological samples, MRI data, speech data, and data that have a higher risk of re-identification or are particularly sensitive (i.e., more personal information than data found through open or registered access). This level of access requires submission of a project description approved by the investigator's Institutional Review Board (IRB) for review by CAPTURE ALS's Tissue and Data Committee, and C-BIG Repository's Tissue and Data Committee.

Researchers using the CAPTURE ALS dataset in C-BIG Repository are encouraged to return their results to CAPTURE ALS after publication, and to disseminate study results through peer-reviewed journal publications, posters, and conference presentations. Researchers requesting CAPTURE ALS registered and controlled access data are required to recognize the CAPTURE ALS database in published work.

The CAPTURE ALS PPAC contributes to dissemination indirectly through its input on foundational documentation such as the CAPTURE ALS Charter, focus on patient-centered study processes, and response to study needs and challenges as identified by the CAPTURE ALS Executive Committee. Direct contributions include input on participant engagement webinars and co-authorship on a publication outlining lessons learned from CAPTURE ALS' active participant engagement strategy [64].

## Limitations

One of the challenges is the recruitment of a heterogeneous patient population and also the constraints posed by the longitudinal design. Due to the number of assessments included in the CAPTURE ALS protocol, recruitment can be challenging as people living with ALS may not have the time or energy for lengthy visits. To minimize participant fatigue and burden, the following adjustments are permitted: 1) screening and baseline visits may be combined, and 2) assessments can be carried over to an alternative day within window limits.

The MRI requirement can result in a selection bias towards individuals with milder disabilities and slower progression. The MRI is not mandatory after the baseline visit, allowing patients to continue their participation even if they can no longer tolerate the MRI scan due to disease progression so that data and biofluid sample collection can continue.

Other limitations include the likelihood of low numbers of participants with diagnoses of PLS, PMA, or presymptomatic cases. To some extent, the use of the El Escorial criteria to define ALS cases may contribute to this limitation. More recently developed criteria, such as the Gold Coast criteria, may broaden the phenotypic range of ALS in future studies and can identify ALS patients with higher sensitivity [65]. CAPTURE ALS remains important to providing comprehensive data regarding the phenotype of MNDs and may guide future dedicated studies to these less common MND presentations. The data collected in this study can be combined with that of other initiatives to assist in meeting thresholds required for meaningful analyses.

The CAPTURE ALS protocol does not include a peripheral electrophysiological measure or collection and banking of postmortem tissue. These could not be integrated into the protocol at launch due to funding restrictions.

CAPTURE ALS is among other international projects that seek to better understand ALS by collecting clinical, genetic, biomarker, neurophysiologic, and/or neuroimaging data [66–73]. Each project includes differences in its study questions, methodology, and inclusion criteria. Nonetheless, there would be advantages to being able to aggregate data between studies, in order to generate data with a large number of participants, representing the full breadth of ALS phenotypes and population diversity. Such data coordination efforts are likely to be a massive undertaking and may benefit from support from international ALS organizations to connect respective project coordinators to determine the feasibility of data harmonization.

## Strengths

ALS is a rare disease with significant individual heterogeneity which makes studying the disease exceptionally challenging. Addressing the variability and heterogeneity will have implications for current therapeutic approaches.

Thus, the CAPTURE ALS platform is designed to address the high variability and heterogeneity observed in ALS patients by providing the most comprehensive clinical picture of individuals living with ALS ever attempted in the field. The unique contributions of CAPTURE ALS include its open science framework and the inclusion of a PPAC by comparison to other large ALS platforms such as Answer ALS, AMBRoSIA, and CReATE among others. The CAPTURE ALS multidisciplinary platform also collects longitudinal MRI scans and detailed neurocognitive and speech data, which is unique, and links biosamples to these datasets. Such a comprehensive dataset is of invaluable significance and will also help understand the complex nature of ALS. This extensive data collection could lead to the identification of novel clinical-biological associations and the development of personalized predictive models for ALS.

## Conclusion

CAPTURE ALS will provide researchers with robustly phenotyped biosamples and data for research exploring ALS biology and heterogeneity.

CAPTURE ALS has the benefit of using lessons learned from the preceding CALSNIC platform, which included standardized neuroimaging, neurocognitive and clinical collections across 9 centres. Lastly, the inclusion of a PPAC ensures that CAPTURE ALS is driving its objectives with the voices of people living with ALS.

Using an open science framework, CAPTURE ALS will provide a comprehensive clinical picture of the disease for the research community to better understand and study ALS. Conducting this observational longitudinal study will provide more insight into variability between people living with ALS and will make access to data easier for researchers to pursue initiatives to eliminate ALS.

## Supporting information

**S1 Appendix. Screen Visit/ Onset Information Form.**
(PDF)

**S2 Appendix. Study Status Form.**
(PDF)

**S3 Appendix. Medical Evaluations Form.**
(PDF)

**S4 Appendix. Family History and Genetic Testing Form.**
(PDF)

**S5 Appendix. ALS Functional Rating Scale – Revised (ALSFRS-R) Administration SOP.**
(PDF)

**S6 Appendix. Vital Capacity Form.**
(PDF)

**S7 Appendix. Neurological Evaluations Form.**
(PDF)

**S8 Appendix. Neurocognitive Testing and Patient Reported Outcome Measures SOP.**
(PDF)

**S9 Appendix. Biosample Collection, Processing, Storage, Shipment SOP.**
(PDF)

**S1 Table. Full List of Features Assessed in Neurological Examination.**
(DOCX)

**S2 Table. Details of the Neurocognitive Battery (affected participants).**
(DOCX)

**S3 Table. Details of the Neurocognitive Battery (control participants).**
(DOCX)

**S4 Table. Details of Neurocognitive Testing Battery.**
(DOCX)

**S5 Table. Summary of Biosample Processing.**
(DOCX)

## Acknowledgments

CAPTURE ALS sincerely thanks each participant for contributing their time and commitment to the study, as well as all staff involved in CAPTURE ALS at participating institutions. The consortium would also like to highlight the significant input by members of the PPAC who notably contributed to the overall development of the protocol. We thank them for their continuous guidance and positive contributions to the CAPTURE ALS platform.

## Author contributions

**Conceptualization:** Claire Magnussen, Robert Bowser, Tania Bubela, Samir Das, Nicolas Dupré, Alan Evans, Angela Genge, Wendy S Johnston, Jason Karamchandani, Jasna Kriz, Ekaterina Rogaeva, Janice Robertson, Peter Seres, Fred Tam, David Taylor, Christine Vande Velde, Yana Yunusova, Sanjay Kalra.

**Data curation:** HyungMo Kang, Harpreet Bhinder, Liziane Bouvier, Tanushka Anand, Marie-Noelle Boivin, Julia Chiappini, Samir Das, Avnit Dhanoa, Nicolas Ferry, Yvonne Frater, Simon J Graham, Russell Greiner, Yasser Iturria Medina, Wendy S Johnston, Kelvin E Jones, Clémence Tremblay-Desbiens, Yana Yunusova, Sanjay Kalra.

**Investigation:** Gerald Pfeffer, Agessandro Abrahao, Nicolas Dupré, Angela Genge, Russell Greiner, Yasser Iturria Medina, Wendy S Johnston, Kelvin E Jones, Jason Karamchandani, Geneviève Matte, Ekaterina Rogaeva, Janice Robertson, Yana Yunusova, Lorne Zinman, Sanjay Kalra.

**Methodology:** Claire Magnussen, Shelagh K Genuis, Angela Genge, Simon J Graham, Fred Tam, Christine Vande Velde, Sanjay Kalra.

**Project administration:** Natalie Saunders, Claire Magnussen, HyungMo Kang, Mathieu Blais, Harpreet Bhinder, Gerald Pfeffer, Shelagh K Genuis, Tanushka Anand, Nicolas Dupré, Nicolas Ferry, Yvonne Frater, Angela Genge, Wendy S Johnston, Jason Karamchandani, Jasna Kriz, Westerly Luth, David Taylor, Clémence Tremblay-Desbiens, Sanjay Kalra.

**Resources:** Rida Abou-Haidar, Marie-Noelle Boivin, Samir Das, Nicolas Ferry, Jason Karamchandani, Peter Seres.

**Software:** Rida Abou-Haidar.

**Supervision:** Claire Magnussen, Gerald Pfeffer, Tanushka Anand, Nicolas Dupré, Angela Genge, Simon J Graham, Wendy S Johnston, Jason Karamchandani, Jasna Kriz, Peter Seres, Fred Tam, Sanjay Kalra.

**Writing – original draft:** Claire Magnussen, HyungMo Kang, Harpreet Bhinder, Tanushka Anand, Sanjay Kalra.

**Writing – review & editing:** Natalie Saunders, HyungMo Kang, Mathieu Blais, Gerald Pfeffer, Shelagh K Genuis, Liziane Bouvier, Sanjay Kalra.

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
