## [Decision Letter · Decision Letter 0]

25 May 2025

Dear Dr. Saunders,

Thank you for submitting your manuscript to PLOS ONE. After careful consideration, we feel that it has merit but does not fully meet PLOS ONE’s publication criteria as it currently stands. Therefore, we invite you to submit a revised version of the manuscript that addresses the points raised during the review process.

We look forward to receiving your revised manuscript.

Kind regards,

Belgin Sever, Ph.D.

Academic Editor

PLOS ONE

Journal Requirements:

[The CAPTURE ALS platform has been made possible with the financial support of Health Canada, through the Canada Brain Research Fund, an innovative partnership between the Government of Canada (through Health Canada) and Brain Canada (https://braincanada.ca/), and of the ALS Society of Canada (https://als.ca/), Alnylam Pharmaceuticals (https://www.alnylam.com/), Regeneron Pharmaceuticals (https://www.regeneron.com/) (SK), the Canadian Institutes of Health Research (CIHR) (https://cihr-irsc.gc.ca/e/193.html) (SK), and the Chris Snow/Calgary Flames Foundation (SK). The onboarding of CHUM (Centre hospitalier de l’université de Montreal) has been made possible with financial support of the ALS Society of Quebec (https://sla-quebec.ca/en/) and ALS Action Canada (https://www.alsactioncanada.org/) (GM).].

4. Please note that funding information should not appear in the Acknowledgments section or other areas of your manuscript. We will only publish funding information present in the Funding Statement section of the online submission. Please remove any funding-related text from the manuscript.

6. Please include a caption for Figure 1.

7. Please upload a copy of Figure 1, to which you refer in your text on page 15. If the figure is no longer to be included as part of the submission please remove all reference to it within the text.

Reviewers' comments:

Reviewer's Responses to Questions

**Comments to the Author**

1. Does the manuscript provide a valid rationale for the proposed study, with clearly identified and justified research questions?

Reviewer #1: Yes

Reviewer #2: Yes

2. Is the protocol technically sound and planned in a manner that will lead to a meaningful outcome and allow testing the stated hypotheses?

Reviewer #1: Partly

Reviewer #2: Yes

3. Is the methodology feasible and described in sufficient detail to allow the work to be replicable?

Reviewer #1: Yes

Reviewer #2: Yes

4. Have the authors described where all data underlying the findings will be made available when the study is complete?

Reviewer #1: Yes

Reviewer #2: Yes

5. Is the manuscript presented in an intelligible fashion and written in standard English?

Reviewer #1: Yes

Reviewer #2: Yes

You may also provide optional suggestions and comments to authors that they might find helpful in planning their study.

Reviewer #1: This is a protocol paper of an important study on ALS in Canada.

I have the following concerns/questions.

1. Regarding diagnosis of ALS, what do the authors think about the Gold Coast criteria? The Gold Coast criteria do not necessarily need UMN signs to diagnose ALS; patients with PMA can be diagnosed with ALS. On the other hand, "possible ALS" of the rEE criteria may include PLS but not PMA. Well-defined PMA, which is readily diagnosed as ALS based on the Gold Coast criteria, would be critical to this cohort because it is to address the heterogeneity of ALS. Recruiting ALS patients without UMN signs (i.e., PMA) would also merit the cohort because it follows up patients longitudinally and may reveal the clinical and imaging-based natural course of PMA, e.g., what proporion of patients with PMA later develops UMN signs or imaging abnormalities.

https://doi.org/10.1016/j.clinph.2020.04.005

2. The ALSFRS-R SOP has recently been harmonized between ENCALS/TRICALS and NEALS. Do the authors consider using the harmonized SOP?

https://doi.org/10.1080/21678421.2023.2260832

Reviewer #2: This article describes a long term natural history study of ALS in Canada, as well as a data collection effort from healthy control subjects. This study exists in the context of other, quite similar efforts occuring in the US and in Europe. All of these efforts have more similarities than differences. If, as contended by all groups, the strengths of these effort lie in their large scales and thus potential to identify potentially different ALS genetic groups, other non genetic risk factors, or potentially disease modification targets unique to specific ALS subtypes, then it would behoove all parties to proactively determine a data analysis plan that merged the data in common. All groups including this one plan on open source data, so that a formal plan for this should not be impossible. I recognize that this is not a unique issue to this project, but one that should be raised within all groups. For the purpose of this manuscript, I'd suggest that the authors consider this and decide whether such an aggregation effort is important and necessary. If so, a discussion on how they would propose to proceed would be warranted. If not, this should be discussed and justified.

Other than this, the description of the data collection and recruitment effort is clear and reasonable, and I support its dissemination.

**Do you want your identity to be public for this peer review?** For information about this choice, including consent withdrawal, please see our Privacy Policy

Reviewer #1: No

Reviewer #2: No

---

## [Author Response · Author response to Decision Letter 1]

22 Jul 2025

Thank you for your comments and questions. Please see below our responses.

Journal Requirements:

Thank you for providing the appropriate URLs. Our manuscript was reviewed for style requirements and file naming. It should now meet PLOS ONE’s requirements.

Thank you for letting us know. In the Funding Information section, “Brain Canada” was updated to “Fondation Brain Canada” as searched through the dropdown menu. “CIHR” was also modified to “Canadian Institutes of Health Research” from the dropdown menu.

Please note that the grant numbers for the awards are unavailable. The funding from “Fondation Brain Canada” was through a partnership with Regeneron Pharmaceuticals, Alnylam Pharmaceuticals and ALS Canada. If these three funding partners should not be included under “Funding Information”, please let us know.

The Chris Snow/Calgary Flames Foundation and ALS Quebec/ALS Action Canada were received through philanthropic means.

The funding statement below is the statement that the CAPTURE ALS Executive Committee includes in their charter and what was agreed with the primary funder, Brain Canada. Please also note that the receipt of the awards was a result of many individuals. The initials included in the statement are those who received the funds to their institution, which were then dispersed as subawards to others as applicable. If the initials need to be adjusted or removed to properly reflect the funding, please do let us know. This is the updated version:

The CAPTURE ALS platform has been made possible with the financial support of Health Canada, through the Canada Brain Research Fund, an innovative partnership between the Government of Canada (through Health Canada) and Brain Canada (https://braincanada.ca/), and of the ALS Society of Canada (https://als.ca/), Alnylam Pharmaceuticals (https://www.alnylam.com/), Regeneron Pharmaceuticals (https://www.regeneron.com/), the Canadian Institutes of Health Research (CIHR) (https://cihr-irsc.gc.ca/e/193.html) (SK, YM), and the Chris Snow/Calgary Flames Foundation (SK). The views expressed herein do not necessarily represent the views of the Minister of Health or the Government of Canada. The onboarding of CHUM (Centre hospitalier de l’université de Montreal) has been made possible with financial support of the ALS Society of Quebec (https://sla-quebec.ca/en/) and ALS Action Canada (https://www.alsactioncanada.org/) (GM). The funders had no role in study design, data collection and analysis, decision to publish, or preparation of the manuscript. There was no additional external funding received for this study.

3. Thank you for stating in your Funding Statement.

The funding statement has been updated as indicated in #2 above, and is included in our cover letter. The online submission form was updated as well to ensure the correct statement is received.

4. Please note that funding information should not appear in the Acknowledgments section or other areas of your manuscript. We will only publish funding information present in the Funding Statement section of the online submission. Please remove any funding-related text from the manuscript.

Thank you for letting us know. The funding information from the manuscript has been removed.

Thank you very much for pointing out that this was not clear. As mentioned in the manuscript main text, this is a protocol document and does not include any data from CAPTURE ALS. All data relating to the protocol are included in the manuscript text or in the supplemental materials, and will be available without requiring further contact. Any additional specific information regarding the study procedures and protocols can be directed to the corresponding author from any qualified investigator.

The Data Availability statement was removed from the manuscript, and updated in the submission portal:

Standard operating procedures (SOPs) and selected case report forms are provided in supporting information.

There is no data published in this manuscript, except for recruitment numbers.

The CAPTURE ALS protocol, SOPs, and information on how to access data and biosamples from the CAPTURE ALS database are published on http://captureals.ca. Data and biosamples can be requested through open science access principles through this URL: https://cbigr-open.loris.ca/. Data will start to be accessible at the beginning of Q4 of 2025.

6. Please include a caption for Figure 1.

Any reference to Figure 1 was removed. Therefore, a caption is not included. Apologies for the confusion.

7. Please upload a copy of Figure 1, to which you refer in your text on page 15. If the figure is no longer to be included as part of the submission please remove all reference to it within the text.

Thank you. As noted in #6, there is no reference to Figure 1.

Thank you for noting this. We appreciate the details. The attached URL was referenced, and we confirm that Supporting Information guidelines have been followed in the resubmission.

Thank you for this important comment and consideration. We conducted a review of the references, and there are no cited papers that have since been retracted. We can add that there are 61 scientific peer-reviewed papers out of a total of 73 references. The other references are websites. References have since been added to the list to respond to the reviewer’s comments.

Thank you for sharing the responses to the reviewers’ questions.

Reviewers

Reviewer #1: This is a protocol paper of an important study on ALS in Canada.

I have the following concerns/questions.

1. Regarding diagnosis of ALS, what do the authors think about the Gold Coast criteria? The Gold Coast criteria do not necessarily need UMN signs to diagnose ALS; patients with PMA can be diagnosed with ALS. On the other hand, "possible ALS" of the rEE criteria may include PLS but not PMA. Well-defined PMA, which is readily diagnosed as ALS based on the Gold Coast criteria, would be critical to this cohort because it is to address the heterogeneity of ALS. Recruiting ALS patients without UMN signs (i.e., PMA) would also merit the cohort because it follows up patients longitudinally and may reveal the clinical and imaging-based natural course of PMA, e.g., what proporion of patients with PMA later develops UMN signs or imaging abnormalities

Thank you very much for raising this excellent point. There is growing literature regarding the use of the Gold Coast criteria for ALS diagnosis and research studies. At the time of CAPTURE ALS’s protocol development, this was prior to the original publication of the Gold Coast criteria, so it was not included. We agree with the Reviewer’s excellent points that the use of the Gold Coast criteria may have improved the phenotypic range of ALS cases in our study. We have added a section to the discussion in the section “Limitations” addressing this very important point.

2. The ALSFRS-R SOP has recently been harmonized between ENCALS/TRICALS and NEALS. Do the authors consider using the harmonized SOP?

Thank you for sharing this question with us. One of CAPTURE ALS’ Executive Committee members is an author on the referenced paper and we do appreciate the work and collaboration that was put into this harmonized version. CAPTURE ALS’ ALSFRS-R SOP is based primarily on the ENCALS SOP, except for the breathing items which are rely on the NEALS SOP. In order to maintain data and rater consistency, the current SOP will remain. However, when CAPTURE ALS is ready to expand and enter another iteration to continue data and biosample collection, this harmonized version will be implemented to ensure consistency across all international studies and platforms.

Reviewer #2: This article describes a long term natural history study of ALS in Canada, as well as a data collection effort from healthy control subjects. This study exists in the context of other, quite similar efforts occuring in the US and in Europe. All of these efforts have more similarities than differences. If, as contended by all groups, the strengths of these effort lie in their large scales and thus potential to identify potentially different ALS genetic groups, other non genetic risk factors, or potentially disease modification targets unique to specific ALS subtypes, then it would behoove all parties to proactively determine a data analysis plan that merged the data in common. All groups including this one plan on open source data, so that a formal plan for this should not be impossible. I recognize that this is not a unique issue to this project, but one that should be raised within all groups. For the purpose of this manuscript, I'd suggest that the authors consider this and decide whether such an aggregation effort is important and necessary. If so, a discussion on how they would propose to proceed would be warranted. If not, this should be discussed and justified.

Thank you for raising this excellent point, which emphasizes the importance of collaboration in the global effort to better understand ALS. We agree wholeheartedly that it would be ideal for data to be collected in a way that allows it to be easily compared to that from other projects. In the case of CAPTURE ALS, there is a specific emphasis on deep phenotyping, neuroimaging, and biomarkers from multiple biosample types (DNA, DNA methylation data, RNA, plasma NfL), and longitudinal data collection. Based on the goals of the project, the precise data points and timing of data collection will vary. However, despite the obvious challenges, there is excellent potential in aggregation efforts as suggested by the Reviewer, and we have added material to the discussion in the “Limitations” section to address this.

In hopes of future broad harmonization and interoperability, the protocol was set up with reflection on what other groups and platforms were collecting, and we shared our protocol with others. We consulted others operating other initiatives such as Robert Bowser.

---

## [Decision Letter · Decision Letter 1]

31 Aug 2025

Comprehensive Analysis Platform to Understand, Remedy, and Eliminate Amyotrophic Lateral Sclerosis (CAPTURE ALS): Study protocol for a Canadian multicenter, multimodal, longitudinal observational study.

PONE-D-24-55157R1

Dear Dr. Kang,

We’re pleased to inform you that your manuscript has been judged scientifically suitable for publication and will be formally accepted for publication once it meets all outstanding technical requirements.

Kind regards,

Massimo Filippi

Academic Editor

PLOS ONE

Additional Editor Comments (optional):

Reviewer #1:

Reviewers' comments:

Reviewer's Responses to Questions

**Comments to the Author**

1. Does the manuscript provide a valid rationale for the proposed study, with clearly identified and justified research questions?

Reviewer #1: Yes

2. Is the protocol technically sound and planned in a manner that will lead to a meaningful outcome and allow testing the stated hypotheses?

Reviewer #1: Yes

3. Is the methodology feasible and described in sufficient detail to allow the work to be replicable?

Reviewer #1: Yes

4. Have the authors described where all data underlying the findings will be made available when the study is complete?

Reviewer #1: Yes

5. Is the manuscript presented in an intelligible fashion and written in standard English?

Reviewer #1: Yes

You may also provide optional suggestions and comments to authors that they might find helpful in planning their study.

Reviewer #1: The reviewer 1's comments have been addressed.

For Your Information:

Regarding the reviewer 2's comments, Japan also has a nation-wide study consortium called Japanese Consortium for Amyotrophic Lateral Sclerosis Research (JaCALS).

**Do you want your identity to be public for this peer review?** For information about this choice, including consent withdrawal, please see our Privacy Policy

Reviewer #1: **Yes: ** Koji Fujita

---

## [Editor Report · Acceptance letter]

PONE-D-24-55157R1

PLOS ONE

Dear Dr. Kang,

I'm pleased to inform you that your manuscript has been deemed suitable for publication in PLOS ONE. Congratulations! Your manuscript is now being handed over to our production team.

Kind regards,

on behalf of

Prof. Massimo Filippi

Academic Editor

PLOS ONE